# Position: World Models as an Intermediary between Agents and the Real World

Sherry Yang [1]

## Abstract

Large language model (LLM) agents trained using reinforcement learning has achieved superhuman performance in low-cost environments like games, mathematics, and coding. However, these successes have not translated to complex domains where the cost of interaction is high, such as the physical cost of running robots, the time cost of ML engineering, and the resource cost of scientific experiments. The true bottleneck for achieving the next level of agent performance for these complex and high-cost domains lies in the expense of executing actions to acquire reward signals. To address this gap, this paper argues that we should use world models as an intermediary between agents and the real world. We discuss how world models, viewed as models of dynamics, rewards, and task distributions, can overcome fundamental barriers of high-cost actions such as extreme off-policy learning and sample inefficiency in long-horizon tasks. Moreover, we demonstrate how world models can provide critical and rich learning signals to agents across a broad set of domains, including machine learning engineering, computer use, robotics, and AI for science. Lastly, we identify the challenges of building these world models and propose actionable items along dataset curation, architecture design, scaling, and evaluation of world models.

## 1 Introduction

The recipe for achieving superhuman artificial intelligence (AI) agents is no longer a mystery. When a high-capacity agent parameterized by deep neural networks or Large Language Models (LLMs) is coupled with a low-cost environment, we have consistently observed the emergence of capabilities that exceed human performance. From the strategic mastery of AlphaGo (Silver et al., 2017) to LLMs securing

[1] New York University. Correspondence to: Sherry Yang <sherryyang@nyu.edu>.

*Proceedings of the 43rd International Conference on Machine Learning*, Seoul, South Korea. PMLR 306, 2026. Copyright 2026 by the author(s).

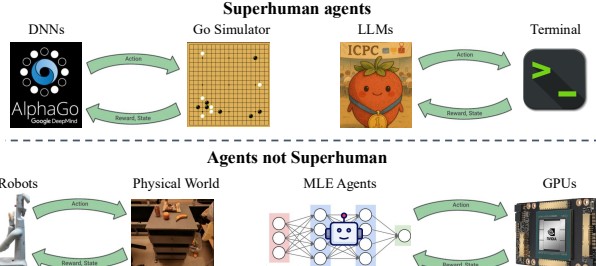

*Figure 1.* **The Cost-Performance Gap.** In low-cost environments with perfect simulators or low-cost execution, we have achieved superhuman agents such as AlphaGo (Silver et al., 2017) and gold-medal winning coding agent (Xu et al., 2025). In high-cost environments such as robotics and ML engineering where execution of each action is constrained by the physical world or can take a long time, we are yet to achieve superhuman agent.

gold medals in the International Mathematical Olympiad (IMO) (Luong et al., 2025) and solving complex coding challenges in ICPC (Xu et al., 2025), the community has demonstrated that given a simulator or a fast feedback loop, we can optimize agents to high levels of proficiency through algorithms such as reinforcement learning (RL) (Sutton et al., 1998).

However, this success has hit a wall in the physical and complex real world (Li et al., 2024), as examples shown in Figure 1. Despite the scaling of base models (Hoffmann et al., 2022) and complex agent scaffolds (Chase, 2022; Significant Gravitas; Wang et al., 2024), we have yet to see LLM agents replace machine learning (ML) engineers (Chan et al., 2024), robots that reliably perform household chores (Zhou et al., 2025), or AI scientists capable of end-to-end discovery (Gottweis et al., 2025). Given the success of RL for games and competition-style math and coding, the bottleneck in achieving these agents is arguably no longer the algorithm (e.g., RL), the architecture (e.g., Transformers), or the optimization method. Rather, the bottleneck lies in the *cost of interaction*. In domains like robotics, an agent is constrained by real-time hardware execution; in ML engineering, a single "action" (training a model) can take days; in scientific discovery, validation can take months. In these high-cost regimes, the trial-and-error approach that fueled successes of existing superhuman agents becomes prohibitively expensive.

To bridge this gap, we should consider decoupling the agent's learning process from the constraints of real-world

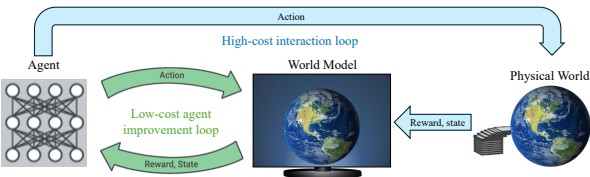

*Figure 2.* **World model as an intermediary between agents and the real world**. In domains where the cost of interactions is high, agents should mostly interact with the world model for training and evaluation, and occasionally interact with the real world for real-world data collection and real-world validation. Most importantly, the real-world interaction data should be used not only to improve the agent but to improve the world model in its ability to predict the next state and reward.

execution. Recently, generative models trained on large-scale real-world data have demonstrated the ability to function as world models—systems that predict future observations, dynamics, and outcomes conditioned on agentic actions in domains such as autonomous driving (Hu et al., 2023), robotics (Yang et al., 2023; Quevedo et al.), code generation (Copet et al., 2025), and social simulacra (Park et al., 2023). By compressing real-world data into a learned simulator, we can transform high-cost, sequential real-world interactions into low-cost, parallelizable queries within a model. These work offer an early indicator of a paradigm shift: instead of relying on brittle, hand-coded simulators or expensive real-world trials, we can accumulate interaction logs to learn a high-fidelity simulator, allowing agents to plan, evaluate, and improve policies.

In this paper, we take the position that **a world model should serve as an intermediary between agents and the real world** for complex, high-cost, or high-stake real-world tasks, as shown in Figure 2. On one hand, the world model can absorb data from high-cost interactions across long time spans. On the other hand, the world model provides a low-cost, high-fidelity environment learned from data for agent training and evaluation. To arrive at this position, we first identify the fundamental difficulties imposed by real-world environments, including the the cost of interaction and the difficulty in stable RL training in the face of extreme off-policyness. This makes online, on-policy RL difficult to achieve. We then discuss how existing alternatives of imitation learning, offline RL, or online RL with a software simulator are not sufficient in achieving the best agentic performance. We emphasize that even in purely digital settings which are supposed to be low cost, privacy and safety constraints often make direct interactions between agents and environments undesirable, further necessitating a world model.

Motivated by these real-world needs, we characterize a world model as a unified system comprising models of (i) dynamics, (ii) rewards, and (iii) task distributions. We illustrate how these functionalities provide not only low-cost interactions and highly realistic task distributions, but also

rich feedback signals and flexible planning horizons for agents. To further illustrate how world models can have a profound impact on real-world applications, we provide an in depth analysis on recent work that utilizes world models to simulate real-world processes, which are then combined with in-context learning, planning, and reinforcement learning (RL) to improve agentic performance in a wide array of applications including robotics, computer use, agentic coding, ML engineering, and AI for science. Lastly, we identify the key challenges in building world models and propose actionable research directions regarding dataset curation, architecture design, scaling, and evaluation to make agents learning from world models for high-cost environments a reality.

## 2 Preliminaries

In this section, we define notations and provide a brief definition of world models in the context of sequential decision making.

**Markov Decision Process (MDPs).** We consider a multi-task, finite-horizon, partially observable Markov Decision Process (POMDP) (Puterman, 2014; Kaelbling et al., 1995), specified by $\mathcal{M} = (S, A, O, G, R, T, \mathcal{E}, H)$, which consists of state, action, observation, and task spaces, reward, transition, and emission functions, and horizon length. A policy $\pi$ interacts with the environment for a task starting from an initial state $g, s_0 \sim G, o_0 \sim \mathcal{E}(s_0)$, producing a distribution $\pi(\cdot|o_t, g)$ over $A$ from which an action $a_t$ is sampled and applied to the environment at each step $t \in [0, H]$. The environment produces a scalar reward $r_t = R(s_t, g)$, and transitions to a new state $s_{t+1} \sim T(s_t, a_t)$ and emits a new observation $o_{t+1} \sim \mathcal{E}(s_{t+1})$.

The goal of an agent can be characterized as maximizing the total expected future reward:

$$\rho(\pi) = \mathbb{E}[R(s_H, g)|s_0, g \sim G, o_t \sim \mathcal{E}(s_t), a_t \sim \pi(o_t, g),$$
$$s_{t+1} \sim T(s_t, a_t) \quad \forall t \in [0, H]], \qquad (1)$$

while performing trial-and-error interactions with the environment.

**Model-Based Learning with a World Model.** In the model-based RL setting (Doya et al., 2002), $T$ and $R$ are unknown but can be estimated from an offline dataset logged from previous interactions $D = \{\tau_i = g, s_0, o_0, a_0, ..., s_H, o_H, r_H\}$. Motivated by characteristics of a real-world system such as image based observations and high control frequencies, the learned model $\hat{T}(\cdot|\mathbf{o}, \mathbf{a})$ can often take a sequence of previous image observations and a sequence of next actions. We define a world model as

$$\mathcal{W} = (\hat{T}, \hat{R}, \hat{G}), \qquad (2)$$

where $\hat{T}$ and $\hat{R}$ are estimated from data similar to model-based RL, and $\hat{G}$ is the real-world task distribution estimated from data. A policy can then trained on task distributions $\hat{G}$ similar to real-world tasks by performing rollout in the

learned model $\mathcal{W}$ as opposed to from the high cost environment, and estimate or optimize the total expected reward through Monte Carlo samples from the world model:

$$\hat{\rho}(\pi) = \mathbb{E}[\hat{R}([o_0, ..., o_H], g)|s_0, g \sim \hat{G}, \mathbf{a} \sim \pi(\mathbf{o}, g),$$
$$\mathbf{o}' \sim \hat{T}(\mathbf{o}, \mathbf{a}), \mathbf{o} = \mathbf{o}']. \tag{3}$$

Various model-based RL (Hafner et al., 2020; Chen et al., 2022; Seo et al., 2022; Micheli et al., 2022; Wu et al., 2022b; Hafner et al., 2023) and planning (Bertoli et al., 2001; Pascanu et al., 2017; Miller et al., 2017) techniques can then be combined with the learned dynamics and reward model to maximize Equation (3).

Nevertheless, existing work in model-based learning focused on learning seperate models and policies per task (Ferns et al., 2004; Achille & Soatto, 2018; Lesort et al., 2018; Castro, 2020; Hafner et al., 2020), as opposed to learning a general world model under a unified observation and action space. This does not unleash the true benefit of world models, since while there can be many tasks and policies, there is only one world in which we live that is governed by the same set of physical laws (Quevedo et al.).

## 3 World Model as an Intermediary

In this section, we motivate the position of world models as an intermediary between agents and the real world by first pointing out the issue of high-cost interactions. We then discuss why existing approaches for handling high-cost interactions fail, and how world models can help.

### 3.1 Difficulty of Learning from High-Cost Interactions

**Time and Sample Efficiency.** As agents are scaled to solve increasingly more complex tasks such as real-world software engineering (Yang et al., 2025a) and ML engineering (Chan et al., 2024), the cost of interaction is prohibitively expensive. For instance, ML engineering benchmarks (Qiang et al., 2025) generally ask an LLM agent to perform an end-to-end ML task, such as image classification. This task requires the agent to load all the training data, building machine learning models, and training the model, and iterate for better ML performance. This can take hours. Similarly, coding in an open-ended environment might require processing data that takes minutes to hours (Shao et al., 2024). This time inefficiency makes the existing approach of RLHF (Castro, 2020) or RL for math (Shao et al., 2024) difficult without an accurate and low-cost reward model.

**Extreme off-policyness.** For efficiency reasons (Thrun, 1992; Kakade, 2003), many RL training frameworks implement an asynchronous distributed setup where multiple "actors" can interact with their own instances of the environment simultaneously, gathering experiences which are then sent to a "learner" for distributed RL gradient updates (Liang et al., 2018; Hoffman et al., 2020). In high-cost environments, rewards for actions may be observed only after long

delays, during which the learner may perform many policy updates. As a result, trajectories are generated by a stale behavior policy $\pi_{\text{old}}$, while gradient updates are applied to a substantially different policy $\pi_\theta$. Policy gradient methods generally rely on importance reweighting,

$$\pi_\theta(a_t \mid o_t, g)/\pi_{\text{old}}(a_t \mid o_t, g), \tag{4}$$

to correct for this mismatch (Schulman et al., 2017; Shao et al., 2024). Even if each update induces only a small change in policy, the effective importance ratio grows (or vanishes) exponentially with the number of intervening updates, leading to extremely high-variance or degenerate gradient estimates. In practice, PPO- or GRPO-style clipping assumes these ratios remain close to one. Under extreme off-policyness, this assumption is violated, causing either unstable updates dominated by a few samples or vanishing gradients when all ratios are clipped. Furthermore, high-cost actions tend to be rare and may be associated with large advantages, amplifying the instability precisely for the most consequential decisions. As a result, asynchronous or delayed-feedback makes RL optimization brittle in high-cost regimes, even with careful tuning.

**Safety and Anonymity.** Action execution time is not the only type of cost that is concerning. Even in digital settings such as computer use where the interaction time may be negligible, direct interaction with the real environment during agent training is still undesirable due to safety and privacy constraints. For instance, in the domain of computer use agents, allowing a policy to interact directly with a live user account poses severe risks, ranging from the manipulation of financial assets to the execution of irreversible malicious behaviors on the internet (Kuntz et al., 2025). Consequently, a world model that functions as a high-fidelity "sandbox" version of the computer and the internet equipped with safety guardrails is desirable. This will allow agents to explore and learn without the risk of catastrophic real-world consequences.

### 3.2 Why Existing Approaches Fail

**Scaling Supervised Learning.** A prevalent approach to circumventing high-cost interactions is to rely exclusively on supervised learning over expert demonstrations (Schaal, 1999). This strategy has been central to scaling Vision-Language-Action (VLA) models in robotics, where vast quantities of teleoperation data are curated to perform supervised finetuning of VLMs (Kim et al., 2024; Black et al., 2024). However, purely supervised agents often lack robustness when facing tasks out-of-distribution (OOD) of the SFT data (Lin et al., 2025). Furthermore, there is a data scalability issue: collecting expert demonstrations is often significantly more expensive and difficult than collecting outcome data for learning a reward function $\hat{R}$ and task distribution $\hat{G}$. As a result, scaling capabilities through RL remains desirable over relying solely on imitated behavior.

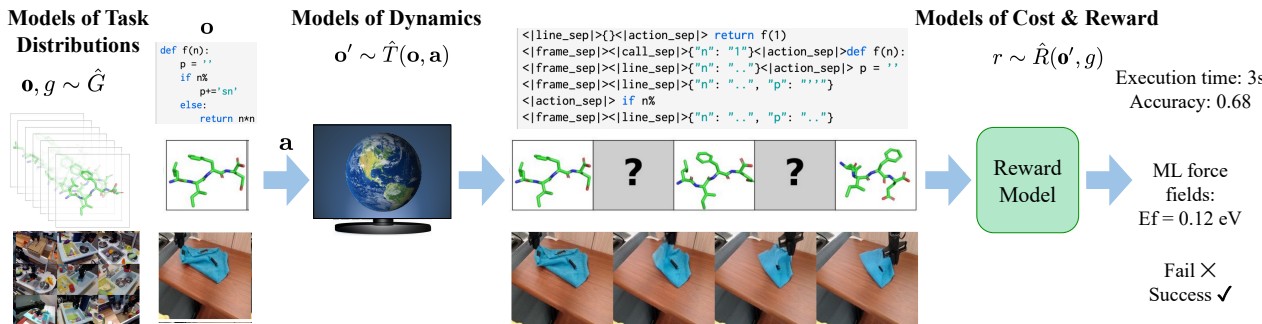

*Figure 3.* **Components of a world model.** A world model can consist of models of realistic task distribution $\hat{G}$, models of dynamics $\hat{T}$, models of reward $\hat{R}$, and models of cost. In code world models (Copet et al., 2025), $\hat{G}, \hat{T}, \hat{R}$ represent realistic coding tasks, execution trace prediction, and performance/cost prediction. In molecular dynamics simulation (Razavian et al., 2012), $\hat{G}, \hat{T}, \hat{R}$ represent realistic initial experimental conditions, dynamics of atom movements, and property prediction such as formation energy. In robotics (Quevedo et al.), $\hat{G}, \hat{T}, \hat{R}$ represent initial configuration of realistic scenes and language tasks, visual dynamics in response to actions, and prediction of success or failure.

**Offline RL.** The paradigm of learning agents from fixed datasets without active interaction has been extensively studied in Offline RL (Lange et al., 2012; Levine et al., 2020), which directly tries to learn an optimal policy only from offline data. While offline RL conceptually addresses the cost of interaction, they introduces severe theoretical limitations, such as the necessity for pessimism in the face of uncertainty (Jin et al., 2021), and empirical difficulties, including optimization instability (Wu et al., 2022a) and sensitivity to hyperparameters (Paine et al., 2020). Another often overlooked limitation is that offline RL formulations are typically task-specific. They rarely leverage unified representations of observations and actions (Yang et al., 2024)—similar to generalist video and text models—from which a general-purpose world model could be learned to support decision-making across a broad spectrum of tasks.

**RL in Simulators.** A traditional alternative to RL in the real world is to train agents within a hand-coded software simulator and attempt zero-shot transfer to the real world (Matas et al., 2018). However, policies trained in such environments frequently suffer from the "sim-to-real" gap. In robotics, this gap occurs as discrepancies between rendered visual features and real-world images as well as difference in simulated and real-world physics (Salvato et al., 2021). In scientific domains, simulated processes often diverge fundamentally from experimental analysis due to unmodeled real-world complexities (Park et al., 2025). These engineered simulators are extensively used by computational science domains but may not approximate experimental processes well, creating a large sim-to-real gap in science, whereas a world model learned directly from experimental data holds the potential to capture the nuanced correlations of the physical world.

### 3.3 Proposal: World Model as an Intermediary

As illustrated in Figure 2, we propose a paradigm where the world model serves as an intermediary between agents and the physical world. In this framework, agents should interact with the real world sparingly, primarily for the purpose of data collection or real-world testing, rather than direct policy optimization. Crucially, real-world data should not be consumed solely to update the agent; instead, it should also be utilized to refine the world model. This improves data utilization of real-world interactions while enabling the agent to generalize better to realistic scenarios that world models can simulate.

In this section, we outline how different components of a world model can be learned from real-world data, and provide examples for each of these components in Figure 3.

**Models of Reward and Dynamics.** Similar to model-based learning, a world model contains a dynamics model $\hat{T}$ and a reward model $\hat{R}$ which are trained on real-world observations and reward outcomes. As shown in Figure 3, by imitating the code execution through predicting the execution outcome without executing the code itself, one can learn a code world model (Copet et al., 2025). Models of dynamics is commonly seen in robotics where a video generation model predicts the visual effect of executing some actions through video prediction conditioned on robot actions (Yang et al., 2023; Guo et al., 2025). This video dynamics model is often combined with a VLM as a reward model to rate the generated videos (Quevedo et al.; Lee et al., 2026). In science domains, researchers have trained generative models to emulate molecular dynamics (Razavian et al., 2012), which hold the promise of being more compute efficient especially in simulating molecular dynamics at a long time-scale. Different from robot world models where the observations are controlled by actions at each discrete step, the setting of molecular dynamics simulation follows

more closely with semi-MDPs (Sutton et al., 1999), where the system evolves under continuous time and discrete intervention (e.g., temperature and pressure applied to the system).

**Models of Cost.** Beyond the dynamics ($\hat{T}$) and rewards ($\hat{R}$) models commonly studied in model-based RL, it is also beneficial for a world model to learn to predict the *cost* of executing actions. This enables the agent to reason about optimal actions under resource constraints settings prior to action execution, such as the time or computational cost for executing each action. For example, an agent might choose to explore a solution space using low-cost, rapid actions before committing to high-latency actions that expand the solution but take hours to complete. Agents that reason about execution cost using the cost information provided by a world model can exhibit superior planning capabilities and resource efficiency. In other words, truly intelligent agents should be able to best balance time spent in learning (gradient updates of parameters), running inference (sampling from the model), and executing actions (which is currently overlooked), and making smart decisions about when to perform which one.

**Models of Task Distribution.** Finally, existing work in model-based learning often overlooks the value of real-world data in defining realistic task distributions ($\hat{G}$), as shown in Figure 3. A world model trained on diverse data inherently captures the distribution of valid and useful tasks. For instance, creating a set of computer use tasks derived from human interaction logs (Shaikh et al., 2025). This can eventually enable agents to be trained on realistic workflows of everyday computer use, as opposed to being trained in simulated environments Xie et al. (2024); Wei et al. (2025) that do not necessarily reflect real-world computer use. Similarly, generative image editing tools trained on internet-scale text-image data can be repurposed to procedurally generate novel manipulation tasks for robots by altering objects in a scene (Quevedo et al.). This suggests that a world model is not merely a simulator of physics, but a generator of curriculum, providing a broad and realistic distribution of tasks for training and evaluating agents.

### 3.4 Iterative World Model and Agent Improvement

As shown in Figure 2, once an agent improves its policy through interacting with a low-cost world model, the agent in principle is better at executing a real-world task. Notably, the world model can serve as a low-cost evaluation proxy before real-world deployment, so that deployment can have better safety and performance guarantees. In testing or deploying the improved agent in the real world, additional data such as the responses from the environment and the actual reward received can be used to iteratively improve the world model and the agent, similar to the classical Dyna algorithm (Sutton, 1991). Compared to the original Dyna

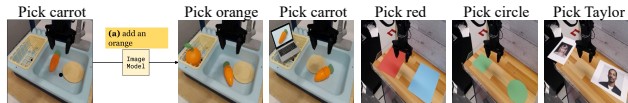

*Figure 4.* **OOD tasks generated by a world model** in WorldGym (Quevedo et al.) through image editing tools such as Nano Banana (Google, 2025). New objects can be inserted to create novel initial observation $o_0$ and new language instructions can be curated to create a new task $g$. These OOD tasks can be used to evaluate, train, and debug policies (e.g., testing if they are better with colors or shapes).

algorithm which was extensively studied and deployed in single-task settings (Zou et al., 2020; Liu & Wang, 2021; Liu et al., 2024), the promise of generalist agents and world models is that they can operate in unified observation spaces (e.g., images) and action spaces (text tokens), resulting in the agent to further generalize across multiple tasks.

## 4 Example Applications

In this section, we discuss concrete applications of world models and highlight how world models unlock key bottlenecks that are otherwise hard to achieve without world models. Furthermore, we point out the challenge in building or using world models for each application domain.

### 4.1 World Models for Robotics

Text-to-video or action-to-video models have recently become of great research interest in robotics (Yang et al., 2023; Guo et al., 2025; Quevedo et al.; Tseng et al., 2025), where text descriptions of a task or robot actions are given to a video generation model as dynamics $\hat{T}$ and VLM as reward $\hat{R}$ to do planning (Pan et al., 2024; Du et al., 2023), policy evaluation (Quevedo et al.; Tseng et al., 2025; Li et al., 2025), and policy improvement (Yang et al., 2023; Guo et al., 2025). 3D world models further extend this paradigm by explicitly modeling geometry and action in physical space (Huang et al., 2026; Team et al., 2025; Yu et al., 2025; Yang et al., 2025c), which can enable 3D consistent interactions with a generated world.

**OOD Task Creation.** One intriguing capabilities of world models for robotics lies in its ability to generate training and evaluation tasks (Guo et al., 2025; Quevedo et al.) and learning curriculums (Wu et al., 2023). As shown in Figure 4, new objects can be added to an existing image through image editing tools such as Nano Banana (Google, 2025), and novel language instructions such as "Pick Taylor Swift" can be given to the policy to test its generalization to OOD objects. Moreover, creating tasks targeting specific abilities of policies such as tasks for selecting different colors and shapes can be useful for debugging limitations of policies.

**Dense Reward and Rich Feedback.** One of the limitations of RL is sparse reward (Hare, 2019), where agents not performing the full task cannot receive reward gradient for progressing learning. With a video generation as dynamics and VLM as reward, the world model $\hat{R}$ can generate not

just 0-1 reward on task success, but also rich feedback on how the policy had failed to complete the task. Such rich feedback can further be utilized by the policy to improve its learning through mechanisms such as self-correction and self-debugging, which are commonly seen in LLMs with dense/process reward models (Kumar et al., 2024; Chen et al., 2023).

**Long-Horizon Planning and RL.** One benefit of the world model is that the action used to plan in the world model does not have to align with the action that needs to physically run on the real robot. For example, previous work has leveraged text as actions to perform long-horizon planning for complex tasks such as rearranging all items on a table into a line (Du et al., 2023) or planning through complex navigation scenarios (Pan et al., 2024). Additionaly, previous research also employed model-based RL to improve policies in a world model, and showed siginificant sample efficiency gains compared to RL on real robots (Zhu et al., 2025a).

**Challenges.** World model for robotics faces the challenge of hallucination (Chu et al., 2024; Guo et al., 2025) and the lack of depth information if the world model is based on video generation (Yang et al., 2025b), Moreover, the generalization abilities of world models are poorly understood (Yang et al., 2024). For instance, a video generation model may fail when presented with novel initial frames or language instructions. It is also unclear whether scaling up data or better architectures are solutions to this generalization challenge. We discuss concrete research directions that can potentially overcome this challenge in Section 5.

### 4.2 World Models for Service Agents

In developing service agents sucuh as airline agents and shopping assistants (Barres et al., 2025; Snell et al., 2022), uncertainty stems from human behavior rather than physical dynamics. Consequently, world models here focus on simulating user responses and latent goals, allowing agents to train and evaluate policies without direct user interaction.

**User Simulation as Dynamics.** The dynamics model $\hat{T}$ acts as a user simulator, generating responses based on agent actions and dialogue history. Unlike early scripted simulators (Jiang et al., 2021), LLMs effectively capture realistic, stochastic, and adversarial behaviors (Park et al., 2023; Barres et al., 2025). This enables agents to train against faithful approximations of real-world conversational dynamics, a setup widely studied in multi-agent contexts (Sun et al., 2024).

**Modeling Task Distributions and Latent Objectives.** Beyond individual turns, world models capture realistic task distributions $\hat{G}$, such as workflows and escalation patterns. Sampling from these distributions allows training on a representative mix of routine and ambiguous interactions,

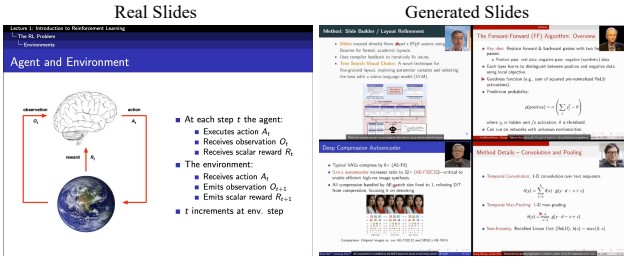

*Figure 5.* **Real and generated computer UI** for slide presentations from Zhu et al. (2025b). The generated application UI and the real application UI can look extremely similar, which provides an opportunity for a world model to faithfully simulate high-fidelity computer use environments, while ensuring proper sandboxing and safety guards.

improving upon static benchmarks that favor clean tasks. World models also address the challenge of latent, delayed objectives (e.g., satisfaction) (Barres et al., 2025) by learning surrogate reward signals $\hat{R}$. By predicting sentiment or resolution outcomes, these signals enable large-scale policy optimization without continuous human feedback.

**Challenges.** A key risk is behavioral collapse under model mis-specification, where agents exploit simulator artifacts or biases (Swift & Leonetti, 2025; Jafferjee et al., 2020). Additionally, modeling long-term adaptation and rare, high-stakes failures remains difficult (Zhu et al., 2024). Addressing these issues requires continual calibration against real logs (Figure 2) and human evaluation to ensure simulated improvements transfer to the real world.

### 4.3 World Models for Computer Use

For computer use agents, direct interaction with real systems during agent training is often undesirable due to safety, privacy, and irreversibility concerns. World models provide a realistic, sandboxed alternative that enables exploration and learning without risking real user accounts or assets.

**UI and Application Simulation as Dynamics.** While existing benchmarks like OSWorld (Xie et al., 2024) and TerminalBench (Merrill et al., 2026) provide environments for training and evaluating computer use agents, they often lack the fidelity of consumer-facing interfaces. A world model $\hat{T}$ addresses this by simulating the evolution of user interfaces (UIs)—either through video generation or by synthesizing executable code (e.g., HTML/JavaScript)—to create a "clone" of the computing environment. In this setup, the model generates interactive elements that visually mimic real applications (as shown in Figure 5) but function within a strict sandbox. This allows an agent to click buttons and execute workflows that feel authentic without triggering external side effects, such as actual bank transactions or data modification (Kuntz et al., 2025).

**Workflow-Level Task Distributions.** World models trained on large-scale interaction logs capture realistic work-

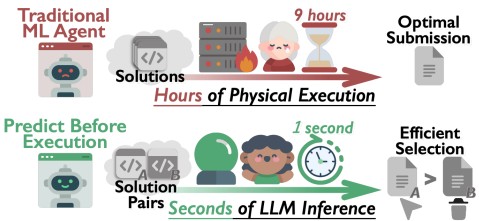

*Figure 6.* **Time-compression with a world model** that predicts experiment outcomes without running the experiment for MLE tasks from Figure 1 in Zheng et al. (2026). Similar time-compressing world models can be applied to other engineering and science domains such as molecular dynamics simulation (Jing et al., 2024).

flow distributions $\hat{G}$ that reflect the complexity of real-world computer use (Shaikh et al., 2025). Unlike limited benchmarks such as WebShop (Yao et al., 2022) or WebArena (Zhou et al., 2023), these models expose agents to long-horizon, tool-rich tasks that reflect true computer use from humans. By training on these learned distributions, agents can master complex workflows in simulation, significantly narrowing the sim-to-real gap.

**Challenges.** Generating high-fidelity simulations presents distinct challenges across modalities. Code-based approaches struggle to reproduce complex site logic without access to proprietary source code, while video-based models require high resolution to ensure precise mouse interaction. Additionally, a fundamental semantic gap remains: because backend mechanics (e.g., payment processing) are approximated rather than executed, agents may still struggle to generalize when moving from the visually identical simulation to the live system.

### 4.4 World Models for Science and Engineering

In science and engineering domains, the primary bottleneck for agentic systems is often the latency and cost of experimentation or high-fidelity simulation. World models offer a mechanism to approximate these processes from real-world data, ideally at a coarser spatial or temporal scales, such as approximating ML engineering processes without executing ML code (Zheng et al., 2026), approximating long time-scale molecular dynamics (Miller et al., 2017), and approximating large systems of particle collision in hadron colliders (Kita et al., 2024).

**Time-Compressed Dynamics.** World models in science and engineering can act as time-compressed approximations of underlying physical or computational processes. For example, an LLM can be trained on real-world ML engineering data to estimate the performance of ML solutions without executing them (Zheng et al., 2026), as shown in Figure 6. In addition, by learning to emulate long-horizon dynamics such as molecular evolution or materials synthesis outcomes, these models can provide predictions at a fraction of the cost of traditional simulations or experiments (Raza-

vian et al., 2012; Jing et al., 2024). This compression can enable rapid iteration over candidate designs or hypotheses.

**Cost and Uncertainty Modeling.** Beyond predicting experimental outcomes, world models can also be used estimate the cost and uncertainty associated with different experimental solutions. This enables agents to reason about trade-offs between exploration and validation under limited time budgets, a critical capability in engineering and scientific settings where failures are expensive and resources are constrained. More importantly, as the agent still has access to the physical environment in our frameing in Figure 2, models of cost and uncertainty can be updated periodically with real-world data, which can lead to better agent performance when trained using more up-to-date world models (Liu et al., 2024).

**Challenges.** A fundamental challenge for scientific and engineering world models is the lack of experimental data to learn such a model of the true scientific or engineering environment, which we provide the call for actions in Section 5.1. Another challenge is to prevent overconfidence in approximate predictions through uncertainty estimation (Seoni et al., 2023). Errors can compound over long horizons, and the world models may fail to extrapolate beyond current observations, which is highly likely in low-data regimes such as engineering and science. Developing uncertainty-aware world models and evaluation protocols remains an open research direction.

## 5 Call to Action

In this section, we propose concrete action items to establish world models as a viable intermediary between agents and the real world, focusing on dataset curation, architecture design, scaling, and evaluation.

### 5.1 Dataset Curation

**Labeling Existing Data** A primary bottleneck in world modeling is the scarcity of action-rich datasets. To address this, we advocate for labeling existing internet-scale text and video data with inferred actions and environmental outcomes. Prior work has successfully utilized image/video captioning models to synthesize text-based actions (Betker et al., 2023; Blattmann et al., 2023) or game actions (Baker et al., 2022). Another promising avenue is leveraging latent actions or skills inferred directly from video data (Edwards et al., 2019; Rybkin et al., 2018; Ye et al., 2022). Furthermore, once sufficient action data is curated to train robust inverse dynamics models (Baker et al., 2022; Venuto et al., 2023), these models can be employed to pseudo-label vast quantities of action-free videos, thereby bootstrapping the dataset for world model training.

**Passive Logging of Environment Information** In domains lacking internet-scale data, such as scientific discovery and engineering, we must enable passive logging

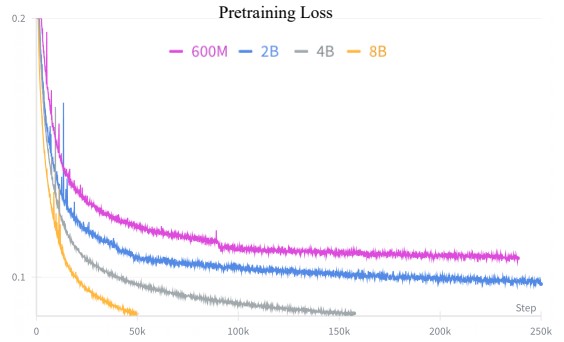

*Figure 7.* **Pretraining loss of video diffusion world models across model sizes.** We train video diffusion models of different sizes as world models and observe that training loss continues to decrease as the model size increases from 600M to 8B parameters, suggesting that we have not yet reached the scaling limit for world models.

mechanisms that automatically record control inputs and system observations (e.g., console outputs, sensor measurements). Since LLMs have demonstrated the ability to mimic human behavior from internet-scale text, they possess the potential to emulate scientific and engineering processes, provided these processes are captured in textual logs. We have already observed that world models can simulate code execution when trained on sufficient execution traces (Copet et al., 2025). We urge the community to begin systematically curating process data from domains like ML engineering, experimental protein and materials synthesis, and automated laboratory workflows. A critical open question remains: *what* should be logged? While capturing every system state is infeasible, we suggest prioritizing the logging of key metrics and observations—akin to loss curves in ML or synthesis conditions in material science—as these features are likely to be the most generalizable.

### 5.2 Model Architecture and Scaling

While Transformer backbones have become the standard for text-based world models, the optimal architecture for world models involving high-dimensional observations (e.g., video, 3D structures) remains an open research question. However, recent advances in image encoding (Peebles & Xie, 2023) and structured data representation (Joshi et al., 2025) offer a promising path for unifying these modalities under Transformer architectures. We call for dedicated research into effective encoders specifically designed for video, 3D, and structural generation within a world model context.

To validate the potential of scaling, we trained latent video diffusion models (Peebles & Xie, 2023; Chen et al., 2024) on action-rich robot and human data using models of varying sizes (600M, 2B, 4B, 8B). As shown in Figure 7, the training loss continues to decrease monotonically as we scale up the model parameters. Based on this observation, we call for future research to further scale both training data and model

size, and to rigorously investigate the scaling laws specific to world models.

### 5.3 Evaluating World Models

**Modularized Evaluation.** A world model comprises three distinct components: $\hat{G}$ (task distribution), $\hat{T}$ (dynamics), and $\hat{R}$ (reward). Since these are trained via maximum likelihood on real-world data, standard generative metrics (e.g., likelihood, FVD, Inception Score) should apply in principle. However, metrics designed purely for perceptual quality may not accurately reflect the *utility* of the world model for downstream planning. A distinct advantage of world models is that their predictions (e.g., images, future structures) are interpretable. We therefore call for the development of modularized evaluation metrics that assess not just the realism, but the physical and functional validity of each component.

**End-to-End Evaluation.** Ultimately, the most robust evaluation of a world model is its downstream utility. If a world model is employed to train an agent via model-based RL, the agent's zero-shot performance in the real world serves as the gold standard metric. While this end-to-end evaluation is the most faithful proxy for utility, it offers limited transparency into specific failure modes. We call for research that leverages world models to instrument agent learning (via RL, planning, or reasoning) and establishes reliable protocols to correlate internal model metrics with end-to-end agent performance.

## 6 Alternative Views

While we advocate for world models as intermediaries, alternative approaches exist. One view suggests scaling real-world interactions directly; however, this is cost-prohibitive and unsafe for high-stakes domains like science or robotics. Another view relies on "reasoning" models to solve tasks zero-shot. We argue that reasoning without the grounding of a world model lacks empirical verification. Ultimately, learned world models provide the only scalable path to bridge the gap between digital reasoning and physical reality.

## 7 Conclusion

Superhuman agents have transformed low-cost domains like coding and games, but they have hit a wall in the physical world where interaction is expensive. In this paper, we proposed world models as the necessary solution to decouple agent learning from real-world constraints. By learning dynamics, rewards, and task distributions from data, world models transform high-cost physical problems into scalable digital ones. We urge the research community to focus on curating action-rich datasets and building these learned intermediaries, unlocking the next frontier of AI in robotics, engineering, and scientific discovery.

# 8  Acknowledgement

We would like to thank Zjin Hu, Ninghao Lu, Ansh Sharma, and Yixiang Sun for sharing scaling curves of pretraining world models. We would like to thank Shenglong Wang and Stratos Efstathiadis for their support for the NYU HPC cluster.

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

# Appendix

## A  Details of Pretraining Scaling

### A.1  Details of the Pretraining Dataset

For the world model scaling experiment, we curate a large dataset of 1M video clips with rich actions including robot manipulation including Open-X Embodiment (**?**), AgiBotWorld-Alpha (contributors, 2024), human manipulation videos including EgoDex (Hoque et al., 2025), H2O (Kwon et al., 2021), ego-centric navigation including Intern Navigation (Contributors, 2025), and all datasets used in Being-H0 (Luo et al., 2025). For datasets that contain robot actions and joint states, we use both during training with a 50% probability of using either as the conditional input to the world model.

### A.2  Details of the Model Architecture

We train a diffusion transformer based model similar to Diffusion Forcing (Chen et al., 2024) and WorldGym (Quevedo et al.), where we use the VAE from Stable Diffusion 3 (Esser et al., 2024) to independently encode $256{\times}256$ image frames into latent space. For different sizes of the models, we vary the number of layers in the transformer and the number of hidden dimensions and attention heads. We pad actions to 256 dimensions as a simple way to combine actions from different robots and human action spaces. We train with a context length of 30 frames.

