# OpenReview forum: "Position: World Models as an Intermediary between Agents and the Real World"
_ICML.cc/2026/Position_Paper_Track — ICML 2026 Position Paper Track regular_

### Official Review · Reviewer_HXvE · 2026-03-02

**Significance:** 2
**Argument Clarity:** 3
**Rating:** 4
**Confidence:** 4

**Questions:**

See the weaknesses.

**Alternative Views Section:**

Yes

**Compliance With Llm Reviewing Policy A Conservative:**

Affirmed.

**Discussion Potential:**

2

**Paper Summary:**

This position paper argues that in complex fields where interaction costs are high (e.g., robot control, machine learning engineering, and scientific discovery) , the World Model should be used as an intermediary layer between agents and the real World. The paper points out that RL has achieved superhuman performance in low-cost environments (games, mathematics, programming), but is limited by interaction costs, extreme off-strategy learning, and sample efficiency problems in high-cost environments; The world model should unify modeling dynamics, rewards, task distribution and execution costs, and transform high-cost physical interactions into low-cost in-model queries This paper systematically expounds the implementation path and unique value of the paradigm in four fields: robotics, service agent, computer use, science and engineering Four specific research agendas are proposed, including data set collation, architecture design, expansion rules and evaluation protocols. Based on the Dyna algorithm, a closed-loop process of collaborative iterative improvement between the world model and the agent is proposed. The contribution of the paper is to provide a systematic and operational paradigm-shift perspective on the “High-cost RL” problem, and to call for community attention to the infrastructure construction of the world model.

**Position:**

Yes

**Position In Title:**

Yes

**Related Work:**

2

**Strengths And Weaknesses:**

Strengths:

- From theoretical difficulties (extreme off-policy, sparse reward), limitations of existing schemes (supervised learning, off-line RL, manual simulator), to world model advantages, the logical chain is complete;
- The suggestions, such as “labeling existing data, “" passive logging,” and “Modular evaluation,” are operable and have practical guiding value to the community.

Weaknesses:

- The paper acknowledges the illusion and generalization challenges of the world model, but lacks quantitative analysis or mitigation strategies for “How model prediction errors accumulate in long-term planning and affect strategy learning”;
- The paper emphasizes that the world model should learn “The universal model under the unified observation/action space”, but it does not clearly define the essential differences between the world model and the existing MBRL methods (such as Dreamer, Transdreamer) , which may lead to conceptual confusion
- The paper calls for end-to-end assessment, but does not propose a quantitative threshold of “What level of fidelity/uncertainty does a world model need to reach to safely replace a real interaction”, weakening the enforceability of the action proposal;

**Support:**

3

---

> ### Author Rebuttal · Authors · 2026-03-31
>
> Thank you for the detailed feedback. Please find our responses below.
>
> > Lack of quantitative analysis or mitigation strategies for "How model prediction errors accumulate in long-term planning"
>
> As suggested, we conducted a quantitative evaluation of temporal degradation in our world model. As shown in the table below, while per-frame quality (PSNR/SSIM) decreases as the horizon extends, the degradation is sub-linear, suggesting that latent representations maintain structural consistency even as pixel-level fidelity drops. We will include this analysis and the discussion in the final paper.
>
> | Metric | MSE (lower) | PSNR (higher) | LPIPS (lower) | SSIM (higher) |
> | :--- | :--- | :--- | :--- | :--- |
> | **24 frames** | 0.008 | 20.51 | 0.14 | 0.79 |
> | **48 frames** | 0.010 | 19.91 | 0.17 | 0.77 |
>
> > Position w.r.t. model-based RL methods (e.g., Dreamer, Transdreamer)
>
> Thank you for pointing out the position with respect to existing model-based RL work. We will be sure to include this discussion in the final paper. In fact, Dreamer and Transdreamer should be a part of the model family choices we consider in this position paper. One difference between the generative approach and latent dynamics modeling approach lies in universality and scale. Traditional MBRL methods like Dreamer are typically "task-specific" or "environment-bound," where the world model and policy are trained jointly on a narrow distribution (e.g., DeepMind Control Suite, Minecraft). Our position explores more on the foundation world model side: models pretrained on massive, heterogeneous datasets (e.g., Open-X) using unified observation/action spaces. Unlike Dreamer, which learns a world model for a task, we focused on world models that act as a generalist infrastructure—a reusable "sandbox" where multiple agents can be trained or evaluated zero-shot.
>
> > Quantitative threshold for fidelity of world model evaluation
>
> We think that a universal "replace/not-replace" threshold is elusive, as risk tolerance is domain-dependent (e.g., high in scientific discovery, low in surgical robotics). However, we can use correlation with the real world as a metrics: the correlation coefficient between policy performance in the World Model and the physical world can serve as a quantitative evaluation for world models. For safety critical settings, we can enforce a high a correlation threshold (e.g., greater than 0.85) and a False Positive Rate (FPR) near zero, meaning the model must never label a catastrophic physical failure as a success. We will update the manuscript to include a protocol for Calibrated Evaluation, where practitioners are recommended to validate the world model on a "held-out physical set" before scaling up in-model training.

---

> > ### Author Rebuttal · Reviewer_HXvE · 2026-04-01
> >
> > Thanks to the author's answers to my questions, almost all my doubts were resolved and I will improve my original score from 3 to 4.

---

### Official Review · Reviewer_ZD38 · 2026-03-12

**Significance:** 3
**Argument Clarity:** 3
**Rating:** 4
**Confidence:** 3

**Questions:**

* The role of the intermediary.
  The paper emphasizes the world model as an intermediary, yet the framework currently appears to treat it primarily as a comprehensive offline simulator. In practice, an intermediary might also need to mediate between simulation and reality. For example, when a policy trained in the world model fails in the real environment (i.e., when a sim-to-real gap appears), what role should the intermediary play? Would it only collect failure data for subsequent offline training, or could it support online correction or adaptation mechanisms?

* Framework cost.
  One of the motivations of the framework is to reduce the cost of agent–environment interaction. However, training and maintaining a large world model may itself be computationally expensive. In what scenarios would the cost of operating such a complex intermediary system exceed the cost of conducting limited exploration directly in the real environment? It would be interesting to understand the potential cost–benefit boundary of this approach.

* Experimental setting in Fig. 7.
  The paper presents a scaling-law curve for the pretraining loss of a video world model in Fig. 7 as preliminary evidence. However, the experimental setup (e.g., dataset, world model architecture, prediction task) is not clearly described. It is therefore unclear whether this experiment directly reflects the proposed framework—particularly the cost model or task distribution model components. As presented, the experiment seems to mainly demonstrate the scalability of the underlying video diffusion model rather than the proposed framework itself.

**Alternative Views Section:**

Yes

**Compliance With Llm Reviewing Policy A Conservative:**

Affirmed.

**Discussion Potential:**

3

**Paper Summary:**

This paper proposes using a comprehensive world model as an intermediary environment in which agents can be trained safely before interacting with the real world. The motivation is to reduce the high costs and risks associated with real-world training. The key contribution of the paper is to extend the traditional notion of a world model beyond a simple dynamics simulator. In particular, the authors propose augmenting the world model with two additional components: a learnable cost model, which predicts resource usage such as time or monetary cost, a task distribution model, which acts as an automated curriculum generator to produce a stream of training tasks. The overall goal is to enable agents to learn from an effectively unlimited stream of realistic tasks in a cost-aware manner before deployment in real-world environments.

**Position:**

Yes

**Position In Title:**

Yes

**Related Work:**

3

**Strengths And Weaknesses:**

Strengths

- Component-level perspective on world models.

  The paper proposes several interesting ideas regarding the internal components of a world model. In particular, the notion of a learnable and generalizable cost model, as well as the use of a task distribution model as a curriculum generator, provides an interesting perspective on how world models could be extended beyond pure environment simulation.

- Creative application scenarios and task-generation ideas.

  The manuscript presents several thought-provoking examples of potential applications, such as time compression for scientific experiments and the generation of tasks from user interaction logs. These examples help illustrate how the proposed framework might be applied in practice.

Weaknesses

- Positioning relative to existing model-based RL literature.

  The central motivation of the paper—that world models can serve as an intermediate training environment before real-world deployment—is already a well-established idea in the model-based reinforcement learning literature. As a result, the novelty of the paper appears to lie more in the proposed extensions to the world model architecture rather than in the general argument itself. Clarifying this positioning would help strengthen the contribution.

- Potential title–content mismatch.

  The title emphasizes the idea of the world model as an “intermediary.” However, much of the paper focuses primarily on the internal components of the world model itself. The intermediary role—such as mediating between simulated and real environments (e.g., sim-to-real transfer or online adaptation)—is discussed less extensively. Expanding this aspect could make the narrative more aligned with the title.

- Limited discussion of curriculum difficulty.

  The paper proposes a task distribution model functioning as a curriculum generator. However, the manuscript does not clearly specify how task difficulty should be measured or evaluated. Without a notion of task difficulty, it is unclear how the curriculum would ensure meaningful progression (e.g., from easier to harder tasks). As a result, the proposed mechanism may resemble task augmentation or sampling unless such metrics are defined more precisely.

**Support:**

3

---

> ### Author Rebuttal · Authors · 2026-03-31
>
> Thank you for the positive feedback! We address your questions as follows.
>
> > Positioning relative to existing model-based RL literature
>
> We clarify that our novelty lies in the "world model as intermediary" concept, rather than specific architectural tweaks. While model-based RL is established, practical impact has been dominated by model-free methods (e.g., RLHF, PPO) due to the difficulty of learning dynamics. We emphasize universal world models—learned from unified observations (images) and action spaces (end-effector poses) across diverse tasks—moving beyond the single-environment focus of classic MBRL.
>
> > Title talks about intermediary but content focuses on inner workings of world models
>
> Thank you for the great suggestion of including more discussions around interactions between agents, world models, and the real world. Reviewer 3UyM also raised a similar question which we addressed by giving concrete workflows when jointly learning a policy and a world model. We had some discussions around Dyna-style algorithms in Section 3.4 which characterizes the RL loop with a learned world model as an intermediary, but we should have included more discussions of the workflow going beyond Dyna, such as using world model to generate high-quality training data, and more concrete use cases of world model as policy evaluator and RL environments. We still do think that emphasizing world models as an intermediary between agents and environments is important and the main takeaway from this work, as this position can have an impact on how people log their data (also store environment executions), and how to design models to emulate environments, especially for high-cost interactive tasks.
>
> > Curriculum learning
>
> Thank you for the great suggestion of using world models to build curriculums for policies. This is definitely something that one can do with a world model that we did not focus on discussing, mostly because curriculum learning is an active area of research on its own. However, we do believe there is a lot of value to using world models to design learning curriculums for policies, especially given that world models can clearly diagnose different failure modes of the policies as shown in Figure 4 of the paper (e.g., policies are good with colors but bad with shapes). We can imagine that world models can create simpler tasks for policies initially and gradually levels up the difficulty as the policy learns the basics. We will be sure to include this discussion in the final paper.
>
> > The role of the intermediary (Sim-to-Real gap)
>
> We envision the world model as an active mediator rather than just a static offline simulator. When a policy fails in the real world, the world model serves two roles: 1) Targeted Data Augmentation: It consumes the real-world failure trace to "patch" its own dynamics, then generates synthetic "near-miss" scenarios to help the policy generalize around that failure point.
> 2) Online Safety Filter: During real-world execution, the world model can run "parallel rollouts" of the policy's proposed actions. If the predicted outcome is high-risk or diverges significantly from the model's confidence interval, the intermediary can trigger a fallback mechanism or request human intervention. We will expand Section 3.4 to include this "Active Intermediary" workflow.
>
> > Framework cost and the cost-benefit boundary
>
> This is a valid concern. The overhead of training a foundation world model is high, but the "break-even" point occurs when: 1_ The marginal cost of real-world data is extreme: E.g., in scientific discovery (weeks per experiment) or robotics (hardware wear-and-tear). 2) Scaling laws take effect: Once a universal world model is pretrained, the cost of fine-tuning it for a new task is orders of magnitude lower than training a policy from scratch in the real world. We will add a discussion on the compute-vs-safety trade-off, noting that while world models are expensive to build, they are a viable path for domains where "exploration" in the real world is physically or ethically impossible.
>
> > Experimental setting in Fig. 7
>
> We apologize for the lack of clarity. Details of the experiment that produced Fig. 7 (e.g., dataset, world model architecture, prediction task) are included in Appendix A. Specifically, we curated a large dataset of 1M video clips with rich actions including robot manipulation including Open-X Embodiment, AgiBotWorld-Alpha, EgoDex, H2O, etc). The model is a latent DiT model trained with diffusion forcing. We will be sure to move these detailed information out of the appendix so these important details are obvious from the main text.

---

> > ### Author Rebuttal · Reviewer_ZD38 · 2026-04-03
> >
> > Thank you for the detailed rebuttal. The response clarifies several of my main concerns. However, some concerns are only partially resolved. In particular, the discussion of the task distribution model as a curriculum generator still lacks a clear notion of task difficulty and how curriculum progression would be defined or evaluated.
> >
> > As a follow-up question, could the authors clarify whether the task distribution model is intended to support an explicit curriculum, and if so, what notion of task difficulty or curriculum ordering they have in mind?

---

### Official Review · Reviewer_8As9 · 2026-03-13

**Significance:** 4
**Argument Clarity:** 4
**Rating:** 6
**Confidence:** 5

**Questions:**

1. How do you propose to mitigate compounding errors during long-horizon rollouts within the world model, especially in low-data and high-stakes domains such as scientific discovery?

**Alternative Views Section:**

Yes

**Compliance With Llm Reviewing Policy A Conservative:**

Affirmed.

**Discussion Potential:**

4

**Final Justification:**

The rebuttal addressed my concerns.

**Paper Summary:**

The paper advocates for using world models (WM) as an intermediary between agents and the real world. The authors first identify *cost of interactions* (in terms of time/compute/safety) as the bottleneck in achieving superhuman performance in many challenging tasks for generalist AI agents (eg robotics, ML engineering, scientific discovery, safety and privacy-constrained digital tasks). The authors then argue that existing approaches for addressing this bottleneck such as imitation learning, offline RL, and online RL in software simulators have failed. They propose to use WMs composed of a generative dynamics model, a reward model, and a task distribution model to create a sandbox environment and simulate real-world task dynamics. Given such a world model, the agent can interact with it to generate trajectories for planning or practice new scenarios before acting in the real world. The authors provide example applications in different domains where WMs can be deployed for improving agents. Call to actions encourage collecting action-rich datasets for training WMs, research into scalable WM architectures, and better evaluation strategies for WMs.

**Position:**

Yes

**Position In Title:**

Yes

**Related Work:**

4

**Strengths And Weaknesses:**

## Strengths

S1. The paper's position is very timely and relevant to the current state of AI research. The authors correctly identify that the current bottleneck in AI agents is the high cost of interactions and that existing approaches fail to overcome this bottleneck. The authors propose generative WMs as the intermediary between the agent and environment and provide support from the literature on why such WMs can be effective in bridging this gap in a range of high-cost interaction domains.

S2. The inclusion of task distribution model and the cost model in the proposed WM framework are valuable for designing better curricula and safety mechanisms for training agents with WMs.

S3. Providing empirical results on scaling properties of diffusion world models (Fig 7) is appreciated and adds merit to the call for action regarding architecture scaling.

## Weaknesses

W1. Although I agree with the authors' argument in the alternative views section that pure model-free control and zero-shot reasoning without grounding are inferior to learned WMs, I am concerned that an alternative view *within* the world modeling community is ignored by the authors; the focus of the paper is on WMs with a generative dynamics model. Non-generative world models have recently shown promising results in reward-based [1, 2] or goal-based [3] planning. If these latent WMs learn an accurate transition model, given a fixed compute budget, they are able to generate many more trajectories compared to generative WMs that can result in better planning. Potentially, one can train language-conditioned decoders on top of these WMs to create OOD tasks too.


## Minor Comments

M1. Page 3, Col 1, Line 123: there are recent works on multi-task model-based RL [1, 2, 4] that are ignored.

M2. Page 2, Col 1, Line 78 (typo): These works offer ...

M3. Page 2, Col 2, Line 107 (typo): A policy can then be trained ...

M4. (Quevedo et al.) does not contain year in citation.

M5. Appendix, Line 721: Open-X Embodiment (?) missing citation.

## References

[1] TD-MPC2: Scalable, Robust World Models for Continuous Control

[2] Learning Massively Multitask World Models for Continuous Control

[3] V-JEPA 2: Self-Supervised Video Models Enable Understanding, Prediction and Planning

[4] Training Agents Inside of Scalable World Models

**Support:**

4

---

> ### Author Rebuttal · Authors · 2026-03-31
>
> Thank you for the positive feedback! We address your questions as follows.
>
> > Alternative view within world modeling
>
> Thanks for this suggestion. Our current draft leans heavily toward generative (pixel/token) world models, and we agree that latent world models (e.g., TD-MPC2, V-JEPA) and 3D-grounded models are essential to this discussion. Latent models often provide superior efficiency for planning, while 3D-aware models offer better spatial consistency than pure video generation. We will add a dedicated subsection in Section 4 to address these paradigms.
>
> Proposed Revision (Section 4.3):
>
> While generative world models offer intuitive human-interpretable grounding, we recognize that latent-space world models provide a compelling alternative for high-frequency control. By predicting transitions in a compressed feature space, models like TD-MPC2 reduce the computational overhead of pixel-level reconstruction and mitigate compounding errors during long-horizon planning. Furthermore, incorporating 3D-aware inductive biases (e.g., Gaussian Splatting or Neural Radiance Fields) into the world model architecture can provide the spatial stability that 2D generative models often lack. We envision a future "universal" world model that likely hybridizes these approaches—utilizing latent dynamics for trajectory optimization while maintaining a generative head for multi-modal grounding and human verification.
>
> > Mitigating compounding errors during long-horizon rollouts in low-data and high-stakes domains
>
> Thank you for raising this great question. While we do not have a perfect answer for this question at the moment (since mitigating error accumulation in low-data regime is an active area of research), we saw signals that scaling the model and dataset size can greatly improve generalization to OOD settings. In extremely low-data settings, we think performing in-context world modeling could be an interesting direction to adapt pretrained world model to low-data regimes, similar to in-context learning for LLMs with few-shot examples. For instance, if the goal is to predict the visual dynamics from the wrist camera conditioned on videos from the wrist camera, having paired videos from wrist and head cameras in context could potentially improve the adaptation of the world model in low-data settings.
>
> > Minor Comments
>
> Thank you for the close read of the manuscript and for pointing out these issues. We will fix them in the final paper.

---

> > ### Author Rebuttal · Reviewer_8As9 · 2026-04-01
> >
> > I thank the authors for their response. My concerns have been addressed and I increased my score.

---

### Official Review · Reviewer_3UyM · 2026-03-18

**Significance:** 3
**Argument Clarity:** 2
**Rating:** 5
**Confidence:** 3

**Questions:**

1. Building universal world models to mitigate high execution/evaluation cost in real-world applications is sound. However, despite from building such a world model, it remains unclear while also promising how the existing AI system can effectively and efficiently utilize the world model. I hope the paper can have further discussion on interaction or workflow between the actor/policy system and world model.

**Alternative Views Section:**

Yes

**Compliance With Llm Reviewing Policy A Conservative:**

Affirmed.

**Discussion Potential:**

2

**Final Justification:**

My concerns have been addressed and I would increase my score.

**Paper Summary:**

This paper addresses the long-standing bottleneck to ground powerful agentic systems which surpasses human-level performance into vast domain of real-world applications is the high cost of execution and evaluation. The authors claims that world model should serve a critical role to alleviate this gap where the agents majority of interactions are with world model and sparingly interact with real physical world. The authors call for research on data curation, architectures, and evaluation.

**Position:**

Yes

**Position In Title:**

Yes

**Related Work:**

3

**Strengths And Weaknesses:**

Strengths

1. This position paper relates to some of classic concepts with similar motivation under the hood, e.g. Dyna-style iterative updates to model-based RL.
2. Comprehensive use cases such as robotics, service agents, computer use, science give concrete illustration how world model might play a fundamentally critical role.

Weaknesses

1. Sim2real gap for learned, universal world models is still under-addressed. Learned models can still face severe OOD or temporally compounded errors.
2. Although it is an important claim/position to ground practical application in real-world, particularly in agentic era. However, world model itself is a long-standing concept in AI. Further enhanced discussions on research directions to scale up universal world models and practical challenges would strengthen the position of this paper.

**Support:**

3

---

> ### Author Rebuttal · Authors · 2026-03-31
>
> Thank you for the positive feedback on this work! We address your questions as follows:
>
> > OOD or temporally compounded errors of world models
>
> It is true that the world model today doesn’t fully generalize to OOD settings and there is compounding errors, but these issues are mitigated with larger model and dataset size as Section 5.2 of the paper. This position paper calls for more research in reducing OOD and compounding errors. We will clarify this in the final paper.
>
> >  Research directions to scale up universal world models
>
> We have identified a few areas of improvements for universal world models in Section 5 such as data/model scaling and evaluation. One important area that requires further investigation (compared to long-standing model-based learning literature) is how to unify different action spaces across diverse data. We can envision research around action tokenizers, discretizations, and perhaps learned encoder/decoders. We will include these additional discussion in the final paper.
>
> > Additional discussion around interaction between the policy and world model
>
> Thank you for this suggestion. We will include additional discussions on the interaction between policy and world model. For example, during training, a policy takes an initial frame from a world model, predicts a sequence of actions, which are fed to the world model, which predicts a sequence of frames, where the last frame are fed back to the policy to predict the next set of actions. During inference, a world model can be used to search (e.g., best of N) and plan for the best actions given a frame. We will include a diagram to make this workflow easy to understand.

---

> > ### Author Rebuttal · Reviewer_3UyM · 2026-04-06
> >
> > I think the author addressed my major concerning points by adding those additional discussions. I will raise score accordingly!

---

### Decision · Program_Chairs · 2026-04-30

**Decision:**

Accept (regular)

**Comment:**

Overall, I find this position paper proposal to be strong and timely. The domain coverage is good, there are well-defined and actionable research directions presented, and the empirical scaling result provides preliminary quantitative grounding. The reviewers are unanimous in voting to accept this paper.

I do encourage the authors to improve the work for the camera-ready. Specifically, a discussion of latent-space world models should be added. Reviewers 3UyM and ZD38 also requested a more concrete workflow of the training loop and inference time procedure. Finally, an expanded discussion of the connections to curriculum learning as a downstream application would be good to include.